# A Review of the Carcinogenic Potential of Thick Rigid and Thin Flexible Multi-Walled Carbon Nanotubes in the Lung

**DOI:** 10.3390/nano15030168

**Published:** 2025-01-22

**Authors:** Omnia Hosny Mohamed Ahmed, Aya Naiki-Ito, Satoru Takahashi, William T. Alexander, David B. Alexander, Hiroyuki Tsuda

**Affiliations:** 1Nanotoxicology Project, Nagoya City University, Nagoya 467-8603, Japan; omnia.hosny@aswu.edu.eg (O.H.M.A.); william@phar.nagoya-cu.ac.jp (W.T.A.); 2Department of Experimental Pathology and Tumor Biology, Graduate School of Medical Sciences, Nagoya City University, Nagoya 467-8601, Japan; ayaito@med.nagoya-cu.ac.jp (A.N.-I.); sattak@med.nagoya-cu.ac.jp (S.T.); 3Department of Forensic Medicine and Clinical Toxicology, Faculty of Medicine, Aswan University, Aswan 81528, Egypt

**Keywords:** multi-walled carbon nanotubes, classification of MWCNTs by IARC, toxicity and carcinogenicity of MWCNTs

## Abstract

The carcinogenic potential of MWCNTs is not well defined. Currently, IARC has classified MWCNT-7 as a Group 2 B material, possibly carcinogenic to humans, and all other MWCNTs as Group 3 materials, inadequate evidence in experimental animals for their carcinogenicity and not classifiable as to their carcinogenicity to humans. In this review we discuss studies that investigated the lung toxicity of well characterized MWCNTs in mice and rats. Intraperitoneal and intrascrotal injection studies identified rigid MWCNTs as hazardous materials. The assessment of lung toxicity of MWCNTs in short and medium term instillation and inhalation studies were not conclusive; therefore, these studies do not confirm the hazard of MWCNTs. However, two-year carcinogenicity studies indicate that MWCNT-7 and other MWCNTs, both thick rigid MWCNTs and thin flexible MWCNTs, are carcinogenic in test animals. Therefore, the carcinogenicity of MWCNTs in experimental animals should be reassessed.

## 1. Introduction

Carbon nanotubes (CNTs) are cylindrical structures that consist of coaxially arranged graphene sheets. According to the number of these sheets, they can be single-walled (SWCNTs), double-walled (DWCNTs), or multi-walled (MWCNTs). MWCNTs can be divided into two general subtypes, flexible and rigid (Figure 1). MWCNTs with low wall numbers are flexible and can assemble into tangled agglomerates. As the wall number increases, the MWCNTs become more rigid and straight. Different types of CNTs vary in their diameter (10 to 200 nm) and length (a few micrometers to tens of micrometers) [1].

Carbon nanotubes are one of the most promising materials in nanotechnology because of their remarkable mechanical, thermal, chemical, and electrical properties. This has led to the high production of CNTs. Over 14,400 tons of CNTs were produced in 2022 [2], with a market value of approximately 6.63 billion U.S. dollars in 2021 which is expected to increase to approximately 20.3 billion U.S. dollars by 2030 [3]. Large-scale production of CNTs greatly increases the potential of workplace and environmental exposure to airborne CNTs and CNT-induced lung toxicity and other diseases [1,4].

Currently, the toxicity of CNTs remains a matter of debate. Based on the effects of fibers administered to the pleura, Stanton (1972) proposed the Stanton Hypothesis, which states “Durable fibers, perhaps at the extreme ranges of dimension, cause cancer simply because they are fibers and irrespective of their physicochemical nature” [5,6]: also see Harington, 1981 [7]. The conclusive study by Stanton is generally considered to be the study published in 1981 in which the carcinogenicity of 72 types of fibrous material was examined [8]. Pott (1974) using intraperitoneal injection also proposed “The carcinogenesis depends on the shape factor of the dusts” [9]: also see Pott (1980) [10]. Results from the Stanton and Pott studies formed the basis of the conclusion that longer straight fibers were carcinogenic while shorter fibers with smaller diameters were less carcinogenic [5,8,9,11,12,13]. The Stanton and Pott hypotheses were based on studies in which fibers were administered directly into the pleural and intraperitoneal cavities: implantation and injection were used because of the difficulty of obtaining the very large quantity of fibers of specific lengths necessary for long-term inhalation studies. Davis et al. (1986) exposed rats to long and short amosite asbestos fibers by inhalation and intraperitoneal injection [14]. They found that intraperitoneal injection of long fibers induced mesothelioma and that inhalation exposure to these long fibers induced lung tumors and the development of two pleural mesotheliomas. One rat injected with 25 mg of short fibers developed a mesothelioma, but no rats injected with 10 mg of short fibers developed mesotheliomas and no rats exposed by inhalation to short fibers developed lung tumors or mesotheliomas. These results not only agree with the Stanton and Pott studies but also suggest that the Stanton and Pott Hypotheses apply to the lung. McDonald et al. (1989) examined lung tissue from 78 fatal cases of mesothelioma and 78 matched referents that died from causes other than mesothelioma or respiratory disease [15]. They concluded that the risk of mesothelioma was related to the concentration in the lung of amphibole fibers greater than 8 µm in length. Thus, the greater tumorigenic potential of longer durable fibers was also found in humans. More recently, the Stanton and Pott hypotheses became incorporated into the Fiber Pathogenicity Paradigm and has been applied to the hazard posed by fibrous nanoparticles to the pleural mesothelium [16] and lung [17].

Intraperitoneal injection studies have supported the Stanton and Pott hypotheses and the Fiber Pathogenicity Paradigm regarding the association of MWCNT fiber dimension with pleural mesothelial carcinogenicity. Based in part on one intrascrotal injection study and one intraperitoneal injection study in rats, two intraperitoneal injection studies in p53^+/−^ mice, and one inhalation-promotion study in mice, in 2014 IARC classified MWCNT-7 as a Group 2B material (see the IARC preamble p30 [18]), there is sufficient evidence for the carcinogenicity of MWCNT-7 in experimental animals, and MWCNT-7 multi-walled carbon nanotubes are possibly carcinogenic to humans [18,19]. CNTs, other than MWCNT-7, were classified as Group 3 materials; there is limited evidence in experimental animals for the carcinogenicity of two types of multi-walled carbon nanotubes with dimensions similar to MWCNT-7 and inadequate evidence in experimental animals for the carcinogenicity of multi-walled carbon nanotubes other than MWCNT-7. Therefore, multi-walled carbon nanotubes other than MWCNT-7 are not classifiable as to their carcinogenicity to humans [18,19]. In the Lancet article, the study by Takagi et al. (2008), the first intraperitoneal injection study in p53^+/−^ mice [20], was omitted from the reference list. As noted below, this study was criticized for the high dose administered to the mice. The MWCNT carcinogenicity data are also discussed in IARC Monograph 111 p190 [18]. Since the classification of MWNCTs other than MWCNT-7 as Group 3 materials, additional carcinogenicity studies in rats have been published. In the present manuscript, we review the in vivo experimental studies of MWCNTs and compare the results with the current hypotheses on the carcinogenicity of inhaled fibers.

## 2. Carcinogenic Studies Using Direct Administration of MWCNTs to the Mesothelium via Intraperitoneal Injection or Intrascrotal Injection (Table 1)

Takagi et al. (2008) administered 3 mg of Mitsui MWCNT-7 (approximately 100 nm in diameter, 1–20 µm in length), 3 mg fullerene, or 3 mg crocidolite asbestos to p53^+/−^ mice [20]. A total of 14 of 16 mice in the MWCNT-7 group developed mesothelioma, 14 of 18 mice in the crocidolite group developed mesothelioma, and no mice in the fullerene or vehicle control groups developed mesothelioma. However, this study was criticized for the high exposure dose used [21,22]. In 2012, Takagi et al. repeated their study using much lower doses of MWCNT-7, 3 µg, 30 µg, and 300 µg per mouse [23]. A total of 5/20, 17/20, and 19/20 mice developed mesotheliomas in the low, middle, and high dose groups, respectively. While p53 heterozygous mice are significantly more sensitive to tumor induction than wild-type mice [24], these results do suggest that MWCNT-7 acts similarly to non-nanoparticle long fibers and induces mesothelium carcinogenicity.

Sakamoto et al. (2009) administered MWCNT-7 and crocidolite asbestos to p53 intact Fischer 344 rats via a single intrascrotal injection [25]. They administered 0.24 mg MWCNT-7 and 0.47 mg crocidolite per rat. By the end of week 52, six of seven rats administered MWCNT-7 had developed mesothelioma while none of the rats administered crocidolite or vehicle developed mesothelial hyperplasia or mesothelioma. These results also indicate that MWCNT-7 is carcinogenic to the mesothelium. While crocidolite administered rats did not develop mesothelioma, Sakamoto et al. noted that it cannot be concluded that MWCNT-7 is a more powerful carcinogen than crocidolite because at the doses administered the particle dose number of MWCNT-7 was much higher than that of crocidolite.

Nagai et al. (2011) administered four different types of MWCNT to Fischer 344/Brown Norway F1 hybrid rats by intraperitoneal injection [26]; Fischer 344/Brown Norway F1 hybrid rats were reported to not develop spontaneous mesothelioma (stated in the Discussion of Aierken et al., 2014 [27]). The rats were observed for up to 350 days. The diameters and lengths of the different MWCNT types determined by the authors were 49.95 ± 0.63 nm and 5.29 ± 0.12 µm (NT50a), 52.4 ± 0.72 nm and 4.6 ± 0.10 µm (NT50b), and 143.5 ± 1.6 nm and 4.34 ± 0.08 µm (NT145). The authors were unable to determine the diameter and length of the NTtngl, the dimensions supplied by the company were 15 nm in diameter and 3 µm in length (NTtngl). The carcinogenicity of NT115, diameter 116.2 ± 1.6 nm and length 4.88 ± 0.08 µm, was not determined. IARC monograph 111 Table 3.2 p70 indicates that NT50a is MWCNT-7 [18]. Nagai et al. report that NT50a, NT50b, and NT145 induced mesotheliomas, with NT50a and NT50b being stronger carcinogens than NT145. NTtngl and vehicle did not induce any mesotheliomas. This is the first study that directly compared the carcinogenicity of a shorter, thin tangled MWCNT with that of longer, rigid MWCNTs. In vitro studies indicated that the thinner rigid MWCNT, NT50a, pierced the cell membranes of hTERT-immortalized human peritoneal mesothelial cells (HPMCs) and had high cytotoxicity, but the thicker NT145 and NT115 MWCNTs did not pierce the membranes of HPMCs and had low cytotoxicity. NTtngl formed bundled aggregates and did not pierce the membranes of HPMCs and also had low cytotoxicity. They also reported that physiochemical factors other than diameter were not the primary cause of HPMC injury. In addition, they reported that NT50a, NT50b, NT115, and NT145 all exhibited toxicity to macrophages (RAW cells) in vitro, but that NTtngl was not toxic to macrophages. Thus, they concluded that NT50a and NT50b exhibited the highest carcinogenic activity due in part to mesothelial cell cytotoxicity and macrophage cytotoxicity and induction of inflammation, that NT145 was carcinogenic due to its macrophage cytotoxicity and induction of inflammation, and that NTtngl was not carcinogenic due to the lack of mesothelial cell or macrophage cytotoxicity (summarized in Figure 8b [26]). Due to the lack of mesothelioma development in the rats administered 10 mg NTtngl, six rats in this group were observed for up to an additional 2 years [28]. These rats developed granulomas, but none of the rats developed mesotheliomas.

Muller et al. (2009) also reported that intraperitoneal injection of short tangled MWCNTs did not induce mesotheliomas in rats [29]. They administered 2 mg or 20 mg of thin tangled MWCNTs or 2 mg crocidolite asbestos by intraperitoneal injection to male Wistar rats. After 2 years, 9 of 26 rats administered crocidolite developed mesotheliomas, 2 of 50 rats administered 2 mg of MWCNT and 0 of 50 rats administered 20 mg of MWCNT developed mesotheliomas, 3 of 50 rats administered MWCNT with defects developed mesotheliomas, and 1 of 26 rats administered vehicle control developed mesotheliomas. Therefore, mesothelioma development was increased in rats administered crocidolite but not in rats administered MWCNT. Rittinghausen et al. (2014) administered four types of thick rigid MWCNTs to male Wistar rats by intraperitoneal injection [30]. All four types of MWCNT induced the development of mesothelioma. However, the diameters of the different MWCNTs reported in Table 1 of Rittinghausen et al. [30] make the results difficult to interpret: 85 ± 1600 nm (MWCNT A), 62 ± 1710 nm (MWCNT B), 40 ± 1570 nm (MWCNT C), and 37 ± 1450 nm (MWCNT D). Sakamoto et al. (2018) administered seven different types of MWCNTs to male Fischer 344 rats by intraperitoneal injection at a dose of 1 mg/kg body weight [31]. Four of the MWCNTs were straight and acicular in shape and three of the MWCNTs were tangled and formed agglomerates. All four straight MWCNTs induced mesotheliomas at a 100% incidence. None of the rats administered tangled MWCNTs developed MWCNT-induced mesotheliomas; two rats administered the tangled MWCNTs developed spontaneous mesotheliomas. The results reported by Muller et al. (2009) and Sakamoto et al. (2018), and possibly the results reported by Rittinghausen et al. (2014) agree with the results reported by Nagai et al. (2011, 2013) that rigid MWCNT fibers, but not thin tangled MWCNT fibers, were carcinogenic to the mesothelium [26,28,29,30,31].

The findings cited above support the premise that MWCNT fibers act similarly to the fibers used in the Stanton and Pott studies and agree the Fiber Pathogenicity Paradigm regarding carcinogenicity of fibers to the mesothelium; the carcinogenicity of MWCNTs to the mesothelium is associated with their length and diameter with longer thicker fibers being more carcinogenic than shorter thinner fibers. An important factor in the carcinogenicity of thin MWCNTs is that thin MWCNT fibers form agglomerates. Consequently, while the dimensions of the fibers affect their carcinogenicity, the particles formed by thinner fibers are not single fibers, but rather tangled agglomerates, and, as reported by Nagai et al. (2011) [26], these agglomerates do not penetrate the cell membranes of mesothelial cells and are not cytotoxic to the mesothelium. Sakamoto et al. (2018) [31] also found that tangled agglomerates of MWCNTs were enclosed in granulomas in the mesothelium, which prevented them from exerting carcinogenicity. Therefore, unlike the short non-carcinogenic fibers studied by Stanton and Pott and other groups, the carcinogenicity of thin MWCNTs is not associated with the carcinogenicity of individual fibers, but rather with the carcinogenicity of agglomerates formed by multiple fibers.

Another important point is that intrascrotal and intraperitoneal injection of test materials does not identify the risk that these materials present to human health, but rather the hazard that these materials represent (see Rittinghausen et al. and Sakamoto et al. [30,31]). A cancer hazard is an agent capable of causing cancer; a cancer risk is the likelihood that cancer will occur when exposed to a cancer hazard (see the IARC preamble p30 [18]). As noted by Sakamoto et al., hazard identification studies should use the most sensitive and easiest method available, see the Discussion in Reference [31]. Therefore, the intrascrotal and intraperitoneal injection studies cited by IARC identified rigid MWCNTs as hazardous materials but did not evaluate their carcinogenicity in experimental animals. As discussed below, the inhalation exposure study by Sargent et al. (2014) indicated that inhalation exposure to MWCNT-7 could promote carcinogenesis in experimental animals [32]. Additional studies supported the premise that the mechanisms of carcinogenesis in rats exposed to MWCNT-7 by inhalation were also operable in humans exposed to airborne MWCNT-7 [18]. Therefore, IARC concluded that there was sufficient evidence for the carcinogenicity of MWCNT-7 in experimental animals, and classified MWCNT-7 multi-walled carbon nanotubes as possibly carcinogenic to humans (Group 2B).

The intrascrotal and intraperitoneal injection studies cited above identified rigid MWCNTs as hazardous to human health. This hazard is supported by the effect of inhaled asbestos on induction of mesothelioma in humans. Studies on the toxicity of MWCNTs in experimental animals exposed to airborne MWCNTs is discussed in the following sections.

**Table 1 nanomaterials-15-00168-t001:** Intraperitoneal (I.P.) and Intrascrotal (I.S.) Injection Studies.

	Injection	Animal	Observation Period	CNT	Diameter Length	Dose per Animal	Mesothelioma
Takagi et al., 2008 [20]	I.P	p53(+/−) Mice	180 days	Control		0 mg	0
				Fullerene		3 mg	0
				Crocidolite		3 mg	14/18
				MWCNT-7	Figure 1 [20]	3 mg	14/16
Takagi et al., 2012 [23]	I.P.	p53(+/−) Mice	1 year	Control		0 mg	0
				MWCNT-7	Diameter: Range 70–170 nm Average 90 nm Length: Range 1–20 µm Average 2 µm	3 µg 30 µg 300 µg	5/20 17/20 19/20
Sakamoto et al., 2009 [25]	I.S.	Fischer 344 rats	52 weeks	Control		0 mg	0
				Crocidolite		0.47 mg	0
				MWCNT-7	Figure 1 [25]	0.24 mg	6/7
Nagai et al., 2011 [26]	I.P.	Fischer 344/Brown F1 rats	350 days	Control		0 mg	0
				NT50a	49.95 ± 0.63 nm 5.29 ± 0.12 µm	1 mg 10 mg	13/13 43/43
				NT50a without aggregation	49.95 ± 0.63 nm 5.29 ± 0.12 µm	same number of fibers as in the 1 mg NT145 group	12/15
				NT50b	52.4 ± 0.72 nm 4.6 ± 0.10 µm	10 mg	6/6
				NT145	143.5 ± 1.6 nm 4.34 ± 0.08 µm	1 mg 10 mg	5/29 28/30
				NTtngl	15 nm 3 µm	10 mg	0/15
Nagai et al., 2013 [28]	I.P.	Fischer 344/Brown F1 rats	3 years	NTtngl	15 nm 3 µm	10 mg	0/6
Muller et al., 2009 [29]	I.P.	Wistar rats	2 years	Control		0 mg	1/26
				Crocidolite		2 mg	9/26
				MWCNT+	11.3 ± 3.9 nm Length ~0.7 µm	2 mg 20 mg	2/50 0/50
				MWCNT-	11.3 ± 3.9 nm Length ~0.7	20 mg	3/50
Rittinghausen et al., 2014 [30]	I.P.	Wistar rats	24 months	Control		0 mg	1/50
				Amosite Asbestos		1.4 mg	33/50
				MWCNT A	85 ± 1600 nm 8.57 ± 151 µm	0.48 mg 2.39 mg	49/50 45/50
				MWCNT B	62 ± 1710 nm 9.30 ± 1.63 µm	0.96 mg 4.80 mg	46/50 45/50
				MWCNT C	40 ± 1570 nm 10.24 ± 1.64 µm	0.87 mg 4.36 mg	42/50 47/50
				MWCNT D	37 ± 1450 nm 7.91 ± 1.40 µm	1.51 mg 7.54 mg	20/50 35/50
Sakamoto et al., 2018 [31]	I.P.	Fischer 344 rats	52 weeks	Control		0 mg	0/10
				M-CNT	66.8 nm 6.65 µm	1 mg/kg	12/12
				N-CNT	59.2 nm 5.48 µm	1 mg/kg	10/10
				WL-CNT	70.9 nm 7.31 µm	1 mg/kg	15/15
				SD1-CNT	177.4 nm 4.51 µm	1 mg/kg	14/14
				WS-CNT (tangled)	44.5 nm 0.5–2 µm	1 mg/kg	0/14
				SD2-CNT (tangled)	13.5 nm Not Determined	1 mg/kg	1/14
				T-CNT (tangled)	35.8 nm Not Determined	1 mg/kg	1/13

## 3. Short and Medium-Term Studies of Pleural Lesions Induced by Administration of MWCNTs into the Lung (Table 2)

Workplace exposure to CNTs is most likely to be inhalation of airborne material. Several studies reported adverse pulmonary and pleural reactions after pharyngeal, intratracheal instillation, nose-only inhalation, or whole-body inhalation of MWCNTs in rats or mice. Table 2 lists four early studies that investigated the effect of on the pleura of the lung administered MWCNTs. Ryman-Rasmussen (2009) administered MWCNTs to male C57BL/6 mice by nose-only inhalation for 6 h [33]. The MWCNTs had diameters of 30–50 nm and lengths of 0.3–50 µm, as determined by an independent contract laboratory (see Supplementary Table 1 in Ryman-Rasmussen [33]). The exposure concentrations were 1 and 30 mg/m^3^. The mice were sacrificed 1 day, 2 weeks, 6 weeks, and 14 weeks after exposure. They reported that MWCNT fibers reached the subpleural wall and caused macrophage aggregation and subpleural fibrosis on the visceral pleura. However, the visceral pleura is very thin in rats (see Figures 3 and 4 in Ryman-Rasmussen [33]). Therefore, it is necessary to consider the possibility that the interaction of the fibers in the alveoli or penetration of fibers deposited in the alveoli into the visceral mesothelium was the cause of the macrophage aggregation and subpleural fibrosis reported by Ryman-Rasmussen. In addition, injection of short and long MWCNTs into the pleural cavity results in retention of the long fibers in the pleural cavity and the development of inflammation and progressive fibrosis on the parietal pleura [34], and in humans mesothelioma generally develops on the parietal mesothelium [16,35,36]. Also, in humans, the visceral pleura is composed of five layers with the upper mesothelial layer separated from the alveolar epithelium by a submesothelial connective tissue layer, a superficial elastica layer, a connective tissue layer, and a fibroelastic layer [37]. Therefore, the formation of macrophage aggregation and fibrosis on the rat visceral mesothelium in the study by Ryman-Rasmussen is unlikely to be relevant to human disease.

Xu et al. (2012) administered two types of thick rigid MWCNTs (see supplementary Figure 1A in Xu et al.) to male F344 rats by intratracheal instillation once every other day for nine days (five administrations) [38]. The diameters of the fibers were not determined. The length of MWCNT-N was 3.64 ± 2.26 µm and the length of MWCNT-7 was 5.11 ± 2.91 µm. MWCNT-7 was designated MWCNT-M in the Xu et al. study. The administered dose of both MWCNT-N and MWCNT-7 was 1.25 mg per rat. Six hours after the last administration, the rats were killed. They report the presence of the MWCNT fibers in the pleural lavage, indicating movement of the fibers into the pleural cavity, and hyperplastic proliferative lesions of the visceral mesothelium. However, cell proliferation appeared to occur primarily in the lung interstitium (see Figure 2A in Xu et al.). Therefore, given the proximity of the lung alveoli to the pleural cavity in rats (see Figure 2A in Xu et al.), similarly to the study reported by Ryman-Rasmussen et al. (2009), the relevance of visceral mesothelial cell proliferation to human disease is unlikely.

Murphy et al. (2013) administered three types of MWCNTs to female C57BL/6 mice by pharyngeal aspiration [39]: the diameters and lengths of the MWCNTs were 14.84 nm and 1–5 µm (tangled), 25.7 nm and 1–2 µm (short rigid), and 165.02 nm and 36 µm (long rigid), respectively (also see supplementary Figure 3 in Murphy et al. [39]). The dose administered was 25 µg per mouse. The mice were sacrificed 1 week and 6 weeks after administration. They report the development of inflammation and fibrosis on the parietal pleura at 6 weeks in mice administered long but not tangled or short MWCNTs. They also reported that these lesions were not apparent at 1 week. An important difference between the study by Murphy et al. and Ryman-Rasmussen was the use of well-defined MWCNT fibers by Murphy et al. [33,39]. An important difference with the study by Xu et al., in addition to the use of mice by Murphy et al., is the time of the terminal sacrifice, nine days after the initial administration of MWCNTs in the study by Xu et al. and seven days (no parietal lesions) and six weeks (lesion development on the parietal pleura) after administration of MWCNTs in the study by Murphy et al. [38,39]. In support of the findings reported by Murphy et al. [39], a later study by Xu et al. (2014) reported that long rigid MWCNT fibers administered to the lung translocated into the pleural cavity and induced a strong inflammatory reaction and fibrosis and mesothelial proliferative lesions in the parietal pleura [40]. Xu et al. (2014) administered two types of MWCNTs to F344 rats by intratracheal instillation [40]. The MWCNTs had diameters and lengths of 15 nm and 3 µm (MWCNT-S) and 150 nm and 8 µm (MWCNT-L), and were administered at doses of 0.125 mg/rat 13 times over a 24 week period (one administration every 2 weeks) for a total dose of 1.625 mg/rat. Rats were sacrificed 24 h after the final instillation. As noted above, MWCNT-L translocated into the pleural cavity and induced a strong inflammatory reaction and fibrosis and mesothelial proliferative lesions in the parietal pleura. Thus, the studies by Murphy et al. (2013) and Xu et al. (2014) are relevant to human disease and support the relevance of intraperitoneal and intrascrotal injection studies that identified long rigid fibers as cancer hazards with the potential to induce the development of mesothelioma [39,40].

**Table 2 nanomaterials-15-00168-t002:** Short and Medium Term Studies of Pleural Lesions Induced by Administration of MWCNTs into the Lung by Instillation and Pharyngeal Aspiration.

	Route of Administration	Animal	Administration/ Exposure Schedule	Sacrifice	CNT	Diameter Length	Dose per Animal	Pleural Lesions
Ryman-Rasmussen et al., 2009 [33]	Nose-only Inhalation	C57BL/6 Mice	6 h	1 day, 2 weeks, 6 weeks, and 14 weeks after administration	Control		0 mg	None
					Carbon Black		30 mg/m^3^	None
					MWCNT	10–30 nm 0.5–40 µm	1 mg/m^3^ 30 mg/m^3^	None Subpleural fibrosis
Xu et al. (2012) [38]	Intratracheal Instillation	F344 rats	Once every other day for 9 days	6 h after the final administration	Control		0 mg	None
					MWCNT-N	Diameter was not determined Length 3.64 ± 2.26 µm	1.25 mg	Proliferative lesions of the visceral mesothelium.
					MWCNT-M (MWCNT-7)	Diameter was not determined Length 5.11 ± 2.91 µm	1.25 mg	Proliferative lesions of the visceral mesothelium.
Murphy et al. (2013) [39]	Pharyngeal Aspiration	C57BL/6 mice	Single administration	1 and 6 weeks after administration	Control		0 µg	None
					Tangled	14.84 nm 1–5 µm	25 µg	None
					Short Rigid	25.7 nm 1–2 µm	25 µg	None
					Long Rigid	165.02 nm 36 µm	25 µg	Inflammation and fibrosis on the parietal pleura at 6 weeks
Xu et al. (2014) [40]	Intratracheal Instillation	F344 rats	Once every 2 weeks for 24 weeks (13 administrations)	24 h after the final administration	Control		0 mg	None
					MWCNT-short	15 nm 3 µm	1.625 mg	Inflammation and fibrosis and mesothelial proliferative lesions in the parietal pleura.
					MWCNT-long	50 nm 8 µm	1.625 mg	Inflammation and fibrosis and mesothelial proliferative lesions in the parietal pleura.

## 4. Short and Medium-Term Studies of Lung Lesions Induced by the Administration of Thick Rigid MWCNTs into the Lung by Instillation and Pharyngeal Aspiration (Table 3)

As discussed above, studies by Murphy et al. and Xu et al. demonstrated that the administration of rigid MWCNTs to the lung could result in the translocation of the fibers to the pleural cavity and induction of proliferative lesions in the parietal pleura [39,40]. In contrast to the effect of rigid MWCNTs in the pleura, Kobayashi et al. (2010) reported that male Sprague Dawley rats administered MWCNT-7 by a single intratracheal instillation did not induce persistent inflammation or other lung lesions [41]. The dose administered by Kobayashi et al. was 0.04, 0.2, and 1 mg/kg body weight and the rats weighed 288–336 g at the time of instillation [41]. Thus, the amount of MWCNT-7 instilled per rat was approximately 12 µg, 60 µg, and 300 µg. The rats were sacrificed 3 days, 1 week, 1 month, 3 months, and 6 months after instillation. Aiso et al. (2010) administered MWCNT-7 to male F344 rats by a single intratracheal instillation [42]. The doses administered were 40 µg and 160 µg per rat, and sacrificed groups of rats 1, 7, 28, and 91 days after instillation. In contrast to Kobayashi et al. (2010), Aiso et al. reported persistent (up to 91 days) dose-dependent induction of epithelial type II hyperplasia and fibrosis [41,42]. Porter et al. (2010) administered MWCNT-7 to male C57BL/6J mice by pharyngeal aspiration at doses of 10, 20, 40, and 80 µg per mouse [43]. The mice were sacrificed 1, 7, 28, and 56 days after administration. In agreement with Aiso et al. (2010), Porter et al. reported that MWCNT-7 caused inflammation and fibrosis in the mouse lungs [42,43]. They also reported that pulmonary inflammation extended from the pulmonary interstitium to the pleura, and MWCNT-7 fibers penetrated into the visceral pleura, which agrees with the proliferative lesions of the visceral mesothelium reported by Ryman-Rasmussen (2009) and Xu et al. (2012), discussed above [33,38]. However, as noted above, the human visceral pleura has five layers with the upper mesothelial layer separated from the alveolar epithelium by a submesothelial connective tissue layer, a superficial elastica layer, a connective tissue layer, and a fibroelastic layer [37] (compare Figures 3 and 4 in Ryman-Rasmussen [33], Figure 2A in Xu et al. [38], and Figure 12 in Porter et al. [43] with Figure 2.35 in Reference [37]), and in humans, mesothelioma generally develops on the parietal mesothelium [16,35,36]. Therefore, the penetration of fibers into the rodent visceral mesothelium is unlikely to be relevant to human disease. As shown in Table 3, the doses of MWCNT-7 administered by Kobayashi et al., Aiso et al., and Porter et al. (rats weigh approximately ten-fold more than mice) encompassed the same range [41,42,43]. Overall, the study by Kobayashi et al. (2010) indicates that administration of up to 1 mg/kg MWCNT-7 into the lung did not cause persistent inflammation or induce lung lesions while the study by Aiso et al. (2010) indicates that the administration of 0.040 mg/kg and 0.640 mg/kg MWCNT-7 into the lung induced fibrosis in the lung and the study by Porter et al. (2010) indicates that the administration of 20 µg/mouse (approximately equivalent to 200 µg/rat or using the values submitted by Aiso et al. [42] approximately equivalent to 800 µg/kg) induced persistent inflammation and fibrosis in the lung [41,42,43]. However, Porter et al. also reported that pulmonary inflammation and fibrosis were resolved at 56 days in mice administered 40 µg MWCNT-7 (approximately equivalent to 1.6 mg/kg) [43]. Therefore, this result agrees with the finding by Kobayashi et al. that the administration of 1.0 mg/kg MWCNT-7 to Sprague Dawley rats did not induce chronic inflammation or fibrosis in the rat lung [41]. Overall, while these early studies do not clarify the toxicity of MWCNT-7 in the lung, the findings of one study in rats and one study in mice do agree that MWCNT-7 is potentially toxic to the lung.

Morimoto et al. (2012) administered another rigid MWCNT (see Figure 1 in Morimoto et al. [44]) to male Wistar rats by a single intratracheal instillation [44]. The MWCNTs had diameters of 48 nm and lengths of 0.94 µm and the doses administered were 0.2 mg (0.66 mg/kg) and 1 mg (3.3 mg/kg) per rat. The rats were sacrificed 3 days, 1 week, 1 month, 3 months, and 6 months after instillation. The low dose caused transient inflammation in the lung and the high dose caused persistent inflammation. This supports the possibility that rigid MWCNT fibers are potentially toxic to the lungs.

Fenoglio et al. (2012) administered MWCNT fibers with diameters of 9.4 nm and 70 nm and lengths of 0.1–1 µm and 1-3 µm, respectively, to female Wistar rats by a single intratracheal instillation [45]. The dose administered was 2 mg/rat, and the rats weighed approximately 250 g. Therefore, the dose administered was much higher than that administered by either Aiso et al. or Kobayashi et al. [41,42]. The rats were sacrificed 3 days after instillation. Fenoglio reported that the thin MWCNTs were toxic to alveolar macrophages in vitro and inflammatory parameters in the lung were high in rats administered the thin MWCNTs. In contrast, thick MWCNTs were less toxic to alveolar macrophages in vitro and the inflammatory parameters in the lung were close to the control levels in rats administered with the thick MWCNTs. These results appear to differ with those reported by Morimoto et al. (2012). Morimoto et al. reported that MWCNT fibers 48 nm in diameter and 0.94 µm in length induced persistent inflammation [44], while Fenoglio reported that MWCNT fibers 70 nm in diameter and 1–3 µm in length induced very little inflammation 3 days after administration [45]. Thus, the findings by Fenoglio suggest that thicker and more rigid fibers (70 nm in diameter) were not inflammatory while the findings by Morimoto suggest that rigid fibers (48 nm in diameter, also see Figure 1 in Morimoto et al. [44]) were inflammatory. As noted above, this difference could be due to fiber-induced cytotoxicity toward alveolar macrophages. Another important difference in the studies by Fenoglio and Morimoto is the timing of post-administration sacrifice. Fenoglio only assessed acute inflammation, which they observed was high in the rats administered thin MWCNTs, while Morimoto assessed both acute inflammation and persistent inflammation, and they observed that persistent inflammation was present in rats administered relatively thick, rigid MWCNTs (see Figure 1 in Morimoto et al. [44]).

As described in Section 3, Murphy et al. (2013) administered tangled (diameter 14.84 nm, length 1–5 µm), short rigid (diameter 25.7 nm, length 1–2 µm), and long rigid (diameter 165.02 nm, length 36 µm) MWCNTs to female C57BL/6 mice by a single pharyngeal aspiration (25 µg/mouse) [39]. The dose administered was 25 µg per mouse. The mice were sacrificed 1 week and 6 weeks after administration. They reported that the short rigid and tangled MWCNTs did not induce lung lesions. In contrast, at 1 week the long rigid MWCNTs induced both inflammation and fibrosis in the lungs, and fibrosis was still present at 6 weeks. As also described in Section 3, Xu et al. (2014) administered thin tangled MWCNTs (diameter 15 nm, length 3 µm) and thick rigid (diameter 150 nm, length 8 µm) MWCNTs to F344 rats by intratracheal instillation once every 2 weeks for 24 weeks (1.625 mg/rat) [40]. The rats were sacrificed 24 h after the final administration. In contrast to Murphy et al., Xu et al. found that while both MWCNT-S and MWCNT-L induced pulmonary inflammation, and MWCNT-S induced a stronger inflammatory reaction than MWCNT-L [39,40]. Poulsen et al. (2015) administered MWCNT fibers with diameters of 11 nm and 67 nm and lengths of 0.85 µm and 4.05 µm, respectively, to female C57BL6 mice by intratracheal instillation [46]. The doses administered were 18, 54, and 162 µg per mouse. The mice were sacrificed 1, 3, and 28 days after administration. Both MWCNTs induced inflammatory responses and fibrosis, with the thicker longer MWCNT inducing more fibrosis at day 28 post-exposure compared to the thinner shorter MWCNT.

Six of the studies cited above [39,40,42,43,44,46] support the possibility that rigid MWCNTs are toxic to the lung, and two of the studies [41,45] reported that administration of rigid MWCNTs did not induce pulmonary inflammation or toxicity. Thus, while these studies mostly support the possibility that rigid MWCNTs are toxic to the lungs, the contrasting results prevent a clear assessment of lung toxicity of thick, rigid MWCNTs.

## 5. Short and Medium-Term Studies of Lung Lesions Induced by Administration of Thin Flexible MWCNTs into the Lung by Instillation and Pharyngeal Aspiration (Table 3)

Notably, the results reported by Fenoglio et al. (2012), Xu et al. (2014), and Poulsen et al. (2015), discussed above, also suggest the possibility that thin fibers are toxic to the lung, which is in contrast to the findings of studies using intraperitoneal injection (IP) to administer MWCNT fibers to rat; IP administration of thin tangled MWCNTs to rats did not induce lesions in the peritoneal mesothelium (Table 1, discussed in Section 2). Other studies also support the possibility of lung toxicity of thin flexible MWCNTs. Muller et al. (2005) administered thin MWCNTs by intratracheal instillation to female Sprague Dawley rats weighing 200–250 g at doses of 0.5, 2, and 5 mg per rat [47]. The fibers were thin and tangled (see Figure 1 in Muller et al. [47]) with diameters of 9.7 ± 2.1 nm and lengths of 5.9 ± 0.05 µm. MWCNT fibers were also ground to produce fibers with diameters of 11.3 ± 3.9 nm and lengths of 0.7 ± 0.07 µm. Rats were sacrificed on days 3, 15, 28, and 60 post-instillation. At 3 days after instillation, they found elevated levels of lactate dehydrogenase (LDH), an indicator of general cytotoxicity, in the bronchoalveolar lavage fluid (BALF) of rats administered 2 mg MWCNT (BALF was not collected from rats administered 5 mg MWCNT), and elevated levels of LDH in the BALF of rats administered 0.5 and 2 mg ground MWCNT (BALF was not collected from rats administered 5 mg ground MWCNT). At 60 days after instillation they found elevated levels of collagen deposition in the lungs of rats administered 2 and 5 mg MWCNT and in the lungs of rats administered 0.5, 2, and 5 mg ground MWCNT. Ronzani et al. (2012) administered MWCNTs to BALB/c mice by intranasal instillation [48]. The MWCNTs had diameters of 10–15 nm and lengths of 0.1–10 µm. The mice were administered doses of 1.5, 6.25, and 25 µg per mouse. They were sacrificed 24 h after a single instillation of 6.25 µg or 7 days after repeated instillation of 1.5, 6.25, and 25 µg; for repeated instillation, the mice were administered MWCNTs on days 0, 7, and 14. Inflammatory parameters were elevated in the BALF of the mice 24 h after a single administration of 6.25 µg MWCNTs. There was a dose dependent increase in the number of macrophages and polymorphonuclear leukocytes in the BALF and collagen deposition in the lungs of the mice 7 days after repeated administration of the MWCNTs. Poulson et al. (2016) administered 10 types of MWCNTs to female C57BL/6J BomTac mice by a single intratracheal instillation [49]. The MWCNTs were administered at doses of 6, 18, 54 µg per mouse. The mice were sacrificed 1, 28, and 92 days post-instillation. The MWCNTs were divided into three groups: Group I with diameters ranging from 20.5 to 26.38 nm and lengths ranging from 518.9 to 1005 nm, Group II with diameters ranging from 26.73 to 32.55 nm and lengths ranging from 771.3 to 1553 nm, and Group III with diameters ranging from 12.96 to 17.22 nm and lengths ranging from 532.5 to 1604 nm. All ten types of MWCNTs induced inflammation in the lung. However, on day 92, only Group III had elevated neutrophil counts. The authors concluded that surface area was a predictor of pulmonary inflammation and genotoxicity. However, Group III MWCNTs had a significantly smaller diameter than the Group I or II MWCNTs, and consequently, more Group III fibers were administered compared to the Group I and II MWCNTs. Overall, the studies cited above indicate that thin tangled MWCNTs administered into the lungs of test rodents are able to induce pulmonary inflammation and cell and tissue damage.

However, the study by Poulsen et al. (2016) [49] also found that pulmonary inflammation induced by six types of thin MWCNTs was resolved at 90 days after administration. The study by Murphy et al. (2013), described above, also found that administration of thin MWCNTs (diameter 14.84 nm, length 1–5 µm) to female C57BL/6 mice by pharyngeal aspiration did not induce inflammation or pulmonary toxicity [39]. The MWCNTs were administered by a single pharyngeal aspiration at a dose of 25 µg per mouse, and the mice were sacrificed 1 week and 6 weeks after administration. Han et al. (2010) administered MWCNTs with diameters of 31 ± 23 nm and lengths of 20 ± 10 µm (similar to the Group II MWCNTs administered by Poulsen et al., 2016, [49]) by oropharyngeal aspiration to female C57B1 mice [50]. The doses were 20 and 40 µg per mouse. The mice were sacrificed 1 day and 7 days post-administration. Inflammatory parameters were significantly increased at day 1. At day 7, the influx of neutrophils remained high, but total protein, LDH levels, and inflammatory cytokines in the bronchoalveolar lavage had returned to basal levels.

Studies [40,45,46,47,48] support the possibility that thin flexible MWCNTs are toxic to the lung, and studies [39,50] reported that administration of thin flexible MWCNTs did not induce persistent pulmonary inflammation or toxicity. The study by Poulsen et al. (2016) [49], reported that MWCNTs with diameters between 12.96 and 17.22 nm caused persistent inflammation, but that MWCNTs with diameters between 20.5 and 32.55 nm did not cause persistent inflammation. Thus, while these studies mostly support the possibility that thin flexible MWCNTs are toxic to the lung, the contrasting results prevent a clear assessment of the lung toxicity of thin flexible MWCNTs.

**Table 3 nanomaterials-15-00168-t003:** Short and Medium Term Studies of Lung Lesions Induced by Administration of MWCNTs into the Lung by Instillation and Pharyngeal Aspiration.

Reference	Route of Administration	Animal	Administration/ Exposure Schedule	Sacrifice	CNT	Diameter Length	Dose per Animal	Lung Lesions
Kobayashi et al., 2010 [41]	Intratracheal Instillation	Sprague Dawley rats	Single administration	3 days, 1 week, 1 month, 3 months, and 6 months after administration	Control		0 mg	None
					MWCNT-7	median diameter 60 nm median length 1.5 µm However, also see Figure 2 [44]	0.04 mg/kg	Pulmonary inflammation was weak and transient. Neither chronic inflammation or fibrosis was observed at any of the doses administered.
							0.2 mg/kg	
							1.0 mg/kg	
Aiso et al., 2010 [42]	Intratracheal Instillation	F344 rats	Single administration	1, 7, 28, and 91 days after administration	Control		0 µg	None
					MWCNT-7	88 nm 5.0 µm	40 µg (160 µg/kg)	Epithelial type II hyperplasia and fibrosis.
							160 µg (640 µg/kg)	Epithelial type II hyperplasia and fibrosis, and induction of microgranulomas.
Porter et al., 2010 [43]	Pharyngeal Aspiration	C57BL/6J mice	Single administration	1, 7, 28, and 56 days after administration	Control		0 µg	None
					MWCNT-7	Figure 4 [40]	10 µg	Inflammation and fibrosis at days 7 and 28.
							20 µg	Inflammation and fibrosis at days 7, 28, and 56.
							40 µg	Inflammation and fibrosis at days 7 and 28.
							80 µg	Inflammation and fibrosis at days 7, 28, and 56.
Morimoto et al., 2012 [44]	Intratracheal Instillation	Wistar rats	Single administration	3 days, 1 week, 1 month, 3 months, and 6 months after administration	Controls		0 mg	None
					MWCNT	48 nm 0.94 µm	0.66 mg/kg	Transient inflammation
							3.3 mg/kg	Persistent inflammation
Fenoglio et al., 2012 [45]	Intratracheal Instillation	Wistar rats	Single administration	3 days after administration	Control		0 mg	None
					MWCNT 9.4	9.4 nm 0.1–1 µm	2 mg	High inflammatory parameters
					MWCNT 70	70 nm 1–3 µm	2 mg	Inflammatory parameters close to control levels
Murphy et al., 2013 [39]	Pharyngeal Aspiration	C57BL/6 mice	Single administration	1 and 6 weeks after administration	Control		0 µg	None
					Tangled	14.84 nm 1–5 µm	25 µg	0
					Short Rigid	25.7 nm 1–2 µm	25 µg	None
					Long Rigid	165.02 nm 36 µm	25 µg	Inflammation at 1 week returned to basal levels at 6 weeks. Fibrosis was present at 6 weeks.
Xu et al., 2014 [40]	Intratracheal Instillation	F344 rats	Once every 2 weeks for 24 weeks(13 administrations)	24 h after the final administration	Control		0 mg	None
					MWCNT-short	15 nm 3 µm	1.625 mg	MWCNT-short was more toxic to the lung than MWCNT-long.
					MWCNT-long	50 nm 8 µm	1.625 mg	
Poulsen et al., 2015 [46]	Intratracheal Instillation	C57BL/6 mice	Single administration	1, 3, and 28 days after administration	Control		0 µg	None
					CNT small	11 nm 0.85 µm	18 µg	Persistent inflammation
							54 µg	Persistent inflammation
							162 µg	Persistent inflammation and fibrosis
					CNT large	67 nm 4.05 µm	18 µg	Persistent inflammation
							54 µg	Persistent inflammation
							162 µg	Persistent inflammation and fibrosis
Muller et al., 2005 [47]	Intratracheal Instillation	Sprague Dawley rats	Single administration	3, 15, 28, and 60 days after administration	Control		0 mg	None
					MWCNT	9.7 ± 2.1 nm 5.9 ± 0.05 µm	0.5 mg	Collagen deposition was not elevated at 60 days
							2 mg	Elevated levels of collagen deposition at 60 days
							5 mg	Elevated levels of collagen deposition at 60 days
					Ground MWCNT	11.3 ± 3.9 nm 0.7 ± 0.07 µm	0.5 mg	Elevated levels of collagen deposition at 60 days
							2 mg	Elevated levels of collagen deposition at 60 days
							5 mg	Elevated levels of collagen deposition at 60 days
Ronzani et al., 2012 [48]	Intranasal Instillation	BALB/c mice	Single administration	24 h after administration	Control		0 µg	None
					MWCNT	10–15 nm 0.1–10 µm	6.25 µg	Elevated inflammatory parameters 24 h after a single administration.
			Administered on days 0, 7, and 14	7 days after administration (Day 21)	Control		0 µg	None
					MWCNT	10–15 nm 0.1–10 µm	1.5 µg	Dose dependent inflammatory response. Dose dependent collagen deposition.
							6.25 µg	
							25 µg	
Poulsen et al., 2016 [49]	Intratracheal Instillation	C57BL/6J BomTac mice	Single administration	1, 28, and 92 days after administration	Control		0 µg	None
					Carbon Black		162 µg	Increase neutrophil counts on days, 1, 28, 92
					Crocidolite		6 µg	Increase neutrophil counts on days 1, 28
							18 µg	Increase neutrophil counts on days 1, 28, 92
					NRCWE-040 Pristine (Group I)	20.56 ± 6.94 nm 518.9 ± 598	6 µg	None
							18 µg	Increase neutrophil counts on day 1
							54 µg	Increase neutrophil counts on days 1, 28
					NRCWE-041 OH-functionalized (Group I)	26.38 ± 11.08 nm 1005 ± 2948 nm	6 µg	Increase neutrophil counts on day 1
							18 µg	Increase neutrophil counts on day 1
							54 µg	Increase neutrophil counts on days 1, 28
					NRCWE-042 COOH-functionalized (Group I)	20.5 ± 5.32 nm 732.2 ± 971.9 nm	6 µg	Increase neutrophil counts on day 1
							18 µg	Increase neutrophil counts on day 1
							54 µg	Increase neutrophil counts on days 1, 28
					NRCWE-043 Pristine (Group II)	26.73 ± 6.88 nm 771.3 ± 3471 nm	6 µg	None
							18 µg	Increase neutrophil counts on day 1
							54 µg	Increase neutrophil counts on days 1, 28
					NRCWE-044 OH-functionalized (Group II)	32.55 ± 14.4 nm 1330 ± 2454 nm	6 µg	None
							18 µg	Increase neutrophil counts on day 1
							54 µg	Increase neutrophil counts on days 1, 28
					NRCWE-045 COOH-functionalized (Group II	28.07 ± 13.85 nm 1553 ± 2954 nm	6 µg	None
							18 µg	Increase neutrophil counts on day 1
							54 µg	Increase neutrophil counts on day 1
					NRCWE-046 Pristine (Group III)	17.22 ± 5.77 nm 717.2 ± 1214 nm	6 µg	Increase neutrophil counts on day 1
							18 µg	Increase neutrophil counts on days 1, 28
							54 µg	Increase neutrophil counts on days 1, 28, 92
					NRCWE-047 OH-functionalized (Group III)	12.96 ± 4.44 nm 532.5 ± 591.9 nm	6 µg	Increase neutrophil counts on day 1
							18 µg	Increase neutrophil counts on days 1, 28
							54 µg	Increase neutrophil counts on days 1, 28, 92
					NRCWE-048 COOH-functionalized (Group III)	15.08 ± 4.69 nm 1604 ± 5609	6 µg	Increase neutrophil counts on days 1, 28
							18 µg	Increase neutrophil counts on days 1, 28
							54 µg	Increase neutrophil counts on days 1, 28, 92
					NRCWE-049 NH2-functionalized (Group III)	13.85 ± 6.09 nm 731.1 ± 1473	6 µg	Increase neutrophil counts on day 1
							18 µg	Increase neutrophil counts on days 1, 28
							54 µg	Increase neutrophil counts on days 1, 28, 92
Han et al., 2010 [50]	Oropharyngeal Aspiration	C57B1 mice	Single administration	1 and 7 days after administration	Controls		0 µg	None
					MWCNT	31 ± 23 nm 20 ± 10 µm	20 µg	Elevated inflammatory parameters at day 1. Elevated levels of neutrophils in the BALF at day 7, but other inflammatory parameters had returned to control levels by day 7.
							40 µg	

## 6. Short and Medium-Term Studies of Lung Lesions Induced by Administration of MWCNTs, Other than MWCNT-7, by Inhalation (Table 4)

Several short and medium-term studies using inhalation of MWCNTs, other than MWCNT-7, are listed in Table 4. Ellinger-Ziegelbauer and Pauluhn (2009) exposed male Wistar rats to thin MWCNTs by nose-only inhalation for 6 h [51]. The MWCNTs contained cobalt. Pristine MWCNTs contained 0.53% cobalt and cobalt-depleted MWCNTs contained 0.11% cobalt. The MWCNTs were 10–16 nm in diameter, but the lengths were not specified. Pauluhn (2010) determined that the lengths of pristine MWCNTs were approximately 200–300 nm (see Figure 3 [52]). The exposure concentrations of pristine MWCNT were 11 and 241 mg/m^3^ and the exposure concentration of cobalt-depleted MWCNT was 11 mg/m^3^. Rats were sacrificed on days 7, 28, and 90. Dose-dependent pulmonary inflammation was observed in the rats exposed to pristine MWCNT. Inflammation in rats exposed to all three MWCNT: pristine-low, pristine-high, and cobalt-depleted, regressed over time. Residual cobalt had little impact on the inflammatory parameters; although the levels of MWCNT used for this comparison, 11 mg/m^3^, had only mild inflammatory activity. The duration of exposure was much less than in the study by Ma-Hock (2009) [53]; however, the highest concentration used by Ellinger-Ziegelbauer and Pauluhn was approximately 100-fold greater than the highest concentration used by Ma-Hock. Thus, the high exposure levels in the study by Ellinger-Ziegelbauer and Pauluhn and the long exposure period in the study by Ma-Hock would account for the induction of inflammation reported in these two studies. In a later study, Pauluhn (2010) exposed female and male Wistar rats to thin tangled MWCNTs by nose-only inhalation for 6 h/day, 5 days/week, for 13 weeks [52]. The median diameter was approximately 10 nm (see Figure 3 [52]) and the lengths were approximately 200–300 nm (see Figure 3 [52]). The MWCNTs were the same as the pristine MWCNT used in the previous study by Ellinger-Ziegelbauer and Pauluhn (2009) [51]. The exposure concentrations were 0.1, 0.4, 1.5, and 6 mg/m^3^. Notably, the MWCNT aerosols were composed of MWCNT agglomerates rather than individual fibers. The rats were sacrificed at weeks 8 (interim sacrifice), 13 (end of the exposure period), 17, 26, and 39. Clearance from the lung was markedly decreased in rats exposed to 0.4, 1.5, and 6 mg/m^3^ MWCNT aerosols. Pauluhn concluded that 0.1 and 0.4 mg/m^3^ covered the range from minimal to moderate lung overload, with exposure to 0.4 mg/m^3^ representing the transitional range where lung overload-related mechanisms start to become operative. Therefore, Pauluhn concluded that the lung pathologies (primarily inflammation, fibrosis, and bronchioloalveolar hyperplasia, Table 3 [52]) observed in this study were the result of lung overload rather than intrinsic toxic properties of the MWCNTs. However, Kasai and Fukushima (2023) concluded that for rats, whole-body inhalation exposure to 2 mg/m^3^ MWCNT-7 did not result in lung overload [54]. They also calculated the clearance rate of MWCNT-7 fibers from the alveoli in rats in their study as Y = 4.5783 X^−0.061^, with Y being the lung burden in µg and X being the days post exposure. The example given in their analysis is that to reduce a lung burden of 4.6 µg to 2.3 µg would require approximately 100,000 days or approximately 274 years: 4.6 = 4.5783 × (1)^−0.061^ and 2.3 = 4.5783 × (100,000)^−0.061^. Therefore, the low clearance rate of MWCNTs out of the lungs of mice exposed to 0.4, 1.5, and 6 mg/m^3^ MWCNT aerosols reported by Pauluhn (2010) is reasonable [52]. In addition, Pauluhn (2010) also states that while the clearance halftime (t_1/2_) of MWCNT fibers from mice exposed to 0.1 mg/m^3^ MWCNT was markedly lower than the clearance halftime of MWCNT fibers from mice exposed to 0.4, 1.5, and 6 mg/m^3^ MWCNT, they could not determine a precise clearance halftime at the 0.1 mg/m^3^ exposure level [52]. While the lung burden calculations by Kasai and Fukushima (2023) [54] cannot be directly applied to the lung burdens in the Pauluhn et al. (2010) study [52], these calculations do indicate that extremely prolonged times were needed for the clearance of MWCNT-7 fibers out of the lung alveoli. Therefore, the low clearance of MWCNT fibers from the lungs of MWCNT-exposed mice cannot be concluded to have been due to non-physiological levels of MWCNTs in the lungs of these animals. Rather, the low clearance rate of MWCNT fibers from the lung may have been due to the normal physiological interaction of MWCNT fibers with alveolar macrophages. Notably, this does not change the conclusion reached by Pauluhn (2010) that the toxic events resulting from MWCNT fibers in the lungs were due to a low clearance rate of fibers out of the lungs rather than to intrinsic toxicity of the fibers [52].

Mitchell et al. (2007) exposed male C57BL/6 mice to thin MWCNTs by whole-body inhalation for 6 h/day for 7 or 14 consecutive days [55]. The MWCNTs were 10–20 nm in diameter and 5–15 µm in length. The exposure concentrations of the MWCNTs were 0.3, 1, and 5 mg/m^3^. The mice were sacrificed the day after the end of exposure. Mitchell et al. report that these thin, probably tangled, MWCNTs did not cause lung damage. Ma-Hock et al. (2009) exposed Wistar rats to thin MWCNTs by nose-only inhalation for 6 h/day, 5 days/week, for 13 weeks [53]. The MWCNTs had diameters of 5–15 nm and lengths of 0.1–10 µm. The exposure concentrations of the MWCNTs were 0.1, 0.5, and 2.5 mg/m^3^. The rats were sacrificed the day after exposure. They reported that exposure to the thin tangled MWCNTs caused granulomatous inflammation in the lung and inflammation in the nasal cavity, larynx, and trachea. An important difference in the studies by Mitchell and Ma-Hock was the duration of exposure, 14 days in the Mitchell et al. study and 65 days in the Ma-Hock study, and the type of exposure, whole body exposure in the Mitchell et al. study and nose-only exposure in the Ma-Hock study. While Mitchell exposed mice and Ma-Hock exposed rats to MWCNT aerosols, the differences in the duration of exposure and the type of exposure could also have a significant effect on the findings reported by the two studies [53,55].

Ma-Hock et al. (2013) exposed male Wistar rats to MWCNTs by nose-only inhalation for 6 h/day for 5 consecutive days [56]. The MWCNTs were thin and tangled (see Figure 3 in Ma-Hock et al. [56]) with diameters of 15 nm. The lengths of the MWCNTs were not determined. The MWCNTs formed large agglomerates of approximately 20 µm diameter. Rats were exposed to 0.1, 0.5, and 2.5 mg/m^3^ MWCNTs. The rats were sacrificed immediately after the last exposure and 3 days, 21 days, and 24 days after the last exposure. Exposure to 0.1 mg/m^3^ MWCNTs did not have a significant effect on lung inflammatory parameters. Exposure to 0.5 mg/m^3^ MWCNTs resulted in an increase in the inflammatory parameters in the lungs 3 days after the last exposure (day 7). These had returned to close to baseline levels by 24 days after the last exposure (day 28). Exposure to 2.5 mg/m^3^ MWCNTs caused a significantly higher increase in the inflammatory parameters in the lung at day 7 compared to the 0.5 mg/m^3^ exposed group, and the inflammatory parameters remained significantly elevated at day 28. Pothmann et al. (2015) exposed Wistar rats to MWCNTs by nose-only inhalation for 6 h/day, 5 days/week, for 90 days [57]. The MWCNTs were thin and tangled (see Figure 5 in Pothmann et al. [57]) with diameters of 12.1 ± 3.5 nm and lengths of 1069 ± 1102 nm. The rats were exposed to 0.05, 0.25, and 5.0 mg/m^3^ and sacrificed 24 h and 90 days after the last exposure. Similarly to the study by Ma-Hock et al. (2013) [56], exposure to 0.05 mg/m^3^ did not have a significant effect on lung inflammatory parameters; exposure to 0.25 mg/m^3^ caused a mild increase in inflammatory parameters in the lung and these had returned to baseline at 90 days post-exposure; exposure to 5.0 mg/m^3^ caused a significantly higher increase in the inflammatory parameters in the lung compared to the 0.25 mg/m^3^ exposed group and these remained elevated at 90 days post exposure. Kim et al. (2020) exposed male Sprague Dawley rats to MWCNTs by nose-only inhalation for 6 h/day, 5 days/week, for 4 weeks [58]. The MWCNTs were thin and tangled (see Figure 1 in Kim et al. [58]) with diameters of 8–10 nm and lengths of 100–200 µm. The MWCNTs formed large agglomerates with diameters of approximately 2.73 µm and lengths of approximately 37.4 µm. The rats were exposed to 0.257, 1.439, and 4.253 mg/m^3^ and sacrificed 1, 7, and 28 days after the last exposure. Similarly to Ma-Hock and Pothmann, exposure to 0.257 mg/m^3^ did not have a significant effect on lung inflammatory parameters, exposure to 1.439 mg/m^3^ caused an increase in inflammatory parameters, and exposure to 4.253 mg/m^3^ caused a more pronounced increase in inflammatory parameters. At 28 days post exposure, the inflammatory parameters in the group exposed to 1.439 mg/m^3^ had returned to control levels while the inflammatory parameters remained elevated in the group exposed to 4.253 mg/m^3^.

The study by Morimoto et al. (2012), described above, also exposed rats by whole-body inhalation to 0.37 ± 0.18 mg/m^3^ MWCNTs with a diameter of 48 nm and a length of 0.94 µm for 6 h/day, 5 days/week, for 4 weeks [44]. The rats were sacrificed at 3 days, 1 month, and 3 months after the end of exposure. They found that 3 days after the end of exposure, inflammatory parameters were increased, but at 1 month and 3 months post-exposure, inflammatory parameters had returned to basal levels. Importantly, Morimoto et al. noted that the amount of MWCNT delivered to the lung by whole-body inhalation was less than the amount administered to the lung in the 1 mg/rat instillation group. Therefore, the results of their inhalation experiment do not contradict the results of their instillation experiment in which they found that 0.2 mg (0.66 mg/kg) MWCNT-induced transient inflammation and 1 mg (3.3 mg/kg) induced persistent in the lung.

All of the studies cited above, except for the study by Mitchell et al. [54], reported that inhalation exposure to thin MWCNTs at 0.1–0.4 mg/m^3^ and above resulted in measurable inflammatory responses (pulmonary inflammation, granulomatous inflammation, fibrosis) to the inhaled MWCNTs. At lower levels of exposure the inflammation resolved after the end of exposure [44,51,56,57,58], while at higher levels of exposure, the inflammation was persistent [56,57,58]. Thus, these studies agree with the majority of the studies discussed in Section 5 that the administration of thin flexible MWCNTs into the lungs of experimental mice and rats can induce inflammation. Consequently, it is possible that thin flexible MWCNTs and thick rigid MWCNTs other than MWCNT-7 are carcinogenic in experimental animals.

**Table 4 nanomaterials-15-00168-t004:** Lung Lesions Induced by Short and Medium Term Inhalation Exposure to MWCNTs other than MWCNT-7.

Reference	Route of Administration	Animal	Administration/ Exposure Schedule	Sacrifice	CNT	Diameter Length	Dose per Animal	Lung Lesions
Ellinger-Ziegelbauer and Pauluhn 2009 [51]	Nose-only inhalation	Wistar rats	6 h	7, 28, 90 days after the end of exposure	Control		0 mg/m^3^	None
					MWCNT-0.11% cobalt	10–16 nm Lengths not specified	10.7 mg/m^3^	Concentration dependent pulmonary inflammation that regressed over time. Cobalt impurities had a minimal effect on the induction or regression of inflammation.
					MWCNT-0.53% cobalt	10–16 nm (Figure 3 [54]) Figure 3 [54]	11.0 mg/m^3^	
							241.3 mg/m^3^	
Pauluhn 2010 [52]	Nose-only inhalation	Wistar rats	6 h/day 5 days per week 13 weeks	8 weeks (interim sacrifice) 13 weeks (end of exposure) Weeks 17, 26, 39	Control		0 mg/m^3^	None
					MWCNT-0.53% cobalt	10 nm (Figure 3 [54]) 200-300 nm (Figure 3 [54])	0.1 mg/m^3^	Minimal fibrosis.
							0.4 mg/m^3^	Fibrosis and minimal inflammation.
							1.5 mg/m^3^	Fibrosis and mild inflammation. Decreased clearance of MWCNT from the lung
							6 mg/m^3^	Fibrosis, inflammation, and epithelial hyperplasia. Decrease clearance of MWCNT from the lung.
Mitchell et al., 2007 [55]	Whole-body inhalation	C57BL/6 mice	6 h/day 7 or 14 days	1 day after the end of exposure	Control		0 mg/m^3^	None
					MWCNT	10–20 nm 5–15 µm	0.3 mg/m^3^ 1 mg/m^3^ 5 mg/m^3^	No lung lesions developed
Ma-Hock et al., 2009 [53]	Nose-only inhalation	Wistar rats	6 h/day 5 days per week 13 weeks	1 day after the end of exposure	Control		0 mg/m^3^	None
					MWCNT	5–15 nm 0.1–10 µm	0.1 mg/m^3^ 0.5 mg/m^3^ 2.5 mg/m^3^	Dose dependent induction of granulomatous inflammation in the lung. Inflammation in the nasal cavity, larynx, and trachea.
Ma-Hock et al., 2013 [56]	Nose-only inhalation	Wistar rats	6 h/day 5 days	Immediately after the last exposure. 3, 21, and 24 days after the last exposure	Control		0 mg/m^3^	None
					MWCNT	15 nm Lengths not determined	0.1 mg/m^3^	None
							0.5 mg/m^3^	Inflammation at day 7 (3 days after the last exposure). Inflammation was resolved at day 28 (24 days after the last exposure).
							2.5 mg/m^3^	Inflammation at day 7 and at day 28.
Pothmann et al., 2015 [57]	Nose-only inhalation	Wistar rats	6 h/day 5 days per week 90 days	24 h and 90 days after the last exposure	Control		0 mg/m^3^	None
					MWCNT	12.1 ± 3.5 nm 1069 ± 1102 nm	0.05 mg/m^3^	None
							0.25 mg/m^3^	Mild induction of inflammation at 24 h. Inflammation was resolved at 90 days after the last exposure.
							5.0 mg/m^3^	Induction of inflammation at 24 h. Inflammatory parameters remained elevated at 90 days.
Kim et al., 2020 [58]	Nose-only inhalation	Sprague Dawley rats	6 h/day 5 days per week 4 weeks	1, 7, and 28 days after the last exposure.	Control		0 mg/m^3^	None
					MWCNT	8–10 nm 100–200 µm	0.257 mg/m^3^	Non-significant increase in inflammatory parameters.
							1.439 mg/m^3^	Increase in inflammatory parameters at days 1 and 7. Inflammatory parameters returned to base line levels at day 28.
							4.253 mg/m^3^	Increase in inflammatory parameters at days 1, 7, and 28.
Morimoto et al., 2012 [44]	Whole-body inhalation	Wistar rats	6 h/day, 5 days per week 4 weeks	3 days, 1 month, and 3 months after exposure	Control		0 mg/m^3^	None
					MWCNT	48 nm 0.94 µm	0.37 ± 0.18 mg/m^3^	Inflammation 3 days after exposure. Inflammatory parameters returned to basal levels at 1 month and 3 months after exposure.

## 7. Two-Year Carcinogenicity Studies (Table 5)

The study by Sargent et al. (2014) was the first to report that exposure to airborne MWCNTs could be carcinogenic [32]. This study initiated lung carcinogenesis by injecting male B6C3F1 male mice with methylcholanthrene at 10 µg/g body weight. One week after injection, the mice were exposed to 5 mg/m^3^ MWCNT-7 by nose-only inhalation for 5 h/day, 5 days/week, for 3 weeks. The mice were sacrificed 17 months after the end of exposure. They reported that 13/56 mice exposed to air (the control group), 28/54 mice injected with methylcholanthrene, 13/49 mice exposed to MWCNT-7, and 38/42 mice injected with methylcholanthrene and then exposed to MWCNT developed lung tumors. This study was the essential study in the IARC classification of MWCNT-7 as a group 2B material: the intraperitoneal and intrascrotal injection studies identified long rigid MWCNTs as cancer hazards, and the study by Sargent et al. identified exposure to MWCNT-7 as representing a cancer risk.

Kasai et al. (2016) exposed F34/DuCrlCrlj rats to 0.02, 0.2, and 2 mg/m^3^ MWCNT-7 (referred to as MWNT-7) for 6 h/day, 5 days/week, and 104 weeks [59]. They found induction of lung carcinomas in male rats exposed to 0.2 mg/m^3^ and 2 mg/m^3^ and in female rats exposed to 2 mg/m^3^ MWCNT-7. The fibers collected from the inhalation chambers had an average diameter of 92.9–98.2 and an average length of 5.4–5.9 µm. The average diameter of the fibers found in the lungs of the rats was 95.5–109.6 nm and the average length of these fibers was 5.8–5.9 µm. Therefore, there was no fraction of fibers with diameters and lengths significantly different from the average fiber that was associated with lung carcinogenicity. The lung burdens in male rats exposed to 0.02, 0.2, and 2 mg/m^3^ were 0.01, 0.15, and 1.8 mg. Importantly, there was no decrease in lung clearance of MWCNT at the carcinogenic levels, indicating that lung overload was unlikely to be the cause of the carcinogenicity of MWCNT-7. This study also reported mesothelial hyperplasia and focal fibrosis in the parietal pleura, but none of the rats developed mesothelioma. The authors concluded that the lack of mesothelioma development was the relatively low number of fibers in the pleura.

The 2-year inhalation study by Kasai et al. was preceded by a study confirming the performance of the MWCNT aerosol generator, another study that tested measurement of the amount of MWCNT fibers in lung tissue, a 2-week toxicity study, and a 13-week toxicity study. An excellent review of whole-body inhalation exposure to MWCNT-7 is provided by Kasai and Fukushima (2023) [54].

Numano et al. (2019) administered MWCNT-7 to male F344/DuCrlCrj rats by intratracheal instillation at a dose of 125 µg per rat once a week for 12 weeks for a total dose of 1.5 mg per rat [60]. In contrast to Kasai et al. (2016) [59], 18 of 19 rats developed malignant mesothelioma, with the rat that did not develop mesothelioma dying at week 93 from a pituitary tumor. Notably, 17 of the 19 rats that died from mesothelioma developed epithelial type II hyperplasia, suggesting that if these rats had not died from mesothelioma they would have developed lung cancer. A primary difference between the studies by Numano et al. and Kasai et al. is that in the study by Numano et al. the entire dose of MWCNT was administered to the rats by the end of week 12 while in the study by Kasai et al. the entire dose of MWCNT-7 was not administered to the rats until the end of 2 years [59,60]. Consequently, in the Numano et al. study, the number of fibers in the pleural cavity was high enough to induce mesothelioma in the rats during the study period [60]. The findings by Numano et al. also suggest that if the rats in the study by Kasai et al. could have lived for another year, they would also have developed mesothelioma [60]. Hojo et al. (2022) administered MWCNT-7 to F344/DuCrlCrlj rats by intratracheal instillation at doses of 0.125 and 0.5 mg/kg body weight 26 times at intervals of 4 weeks [61]; the final dose per rat was similar to that administered by Numano et al. In the study by Hojo et al. rats developed both lung cancer and mesothelioma: 3 of 29 rats in the low-dose group and 9 of 28 rats in the high-dose group developed lung tumors, and 4 of 29 rats in the low-dose group and 12 of 28 rats in the high dose group developed malignant pleural mesothelioma [61]. Thus, in this study, large amounts of MWCNT-7 were administered early enough for mesothelioma to be induced and to develop, but not kill the rats before the development of lung tumors.

Suzui et al. (2016) administered a thick rigid MWCNT (MWCNT-N) to rats via intratracheal instillation [62]. The MWCNTs were filtered through a sieve with a pore size of 25 µm. The MWCNT-N had diameters within 30–80 nm. The average length of the unfiltered MWCNT-N was 4.2 ± 2.9 µm. The average length of the MWCNT-N in the flow-through fraction was 2.6 ± 1.6 µm. The length of the retained MWCNT-N could not be determined. The MWCNT-N fractions were administered at a dose of 125 µg per rat eight times over a 2 week period. This procedure was called TIPS (trans-tracheal intrapulmonary spraying). The final sacrifice was at week 109. None of the rats in the untreated or vehicle control group developed lung tumors or mesothelioma. In the groups administered unfiltered MWCNT-N (U), flow through MWCNT-N (FT), and retained MWCNT-N (R) 6 of 38 rats developed malignant mesothelioma (3 of 12 [U], 3 of 12 [FT], 0 of 14 [R]) and 14 of 38 rats developed lung tumors (4 of 12 [U], 3 of 12 [FT], and 7 of 14 [R]). These studies indicate that when rigid MWCNT fibers are introduced into the lung they are able to induce lung tumors and malignant mesothelioma, a property shared with asbestos.

Saleh et al. (2020) administered a thick, rigid MWCNT and a thin, tangled MWCNT to male F344/DuCrlCrlj rats by TIPS once a week for 7 weeks (a total of eight administrations) [63]. The MWCNTs were administered at doses of 62.5 and 125 µg per rat for total doses of 0.5 and 1 mg per rat. In solution, the thick rigid MWCNT (MWCNT-A) had an average diameter of 150 ± 43 nm and an average length of 6.39 ± 3.07 µm. In solution, the thin, tangled MWCNT (MWCNT-B) had an average diameter of 7.4 ± 2.7 nm and an average length of 1.04 ± 0.71 µm. SEM images of MWCNT-A and MWCNT-B are shown in Figure 8 of Saleh et al. [63]. The final sacrifice was at week 104 after the first administration of MWCNTs. None of the rats developed mesothelioma, consistent with the slow translocation of large fibers from the lung to the pleural cavity [64]. One rat in the vehicle group developed lung tumors. Five rats in the low-dose MWCNT-A group developed lung tumors and four rats in the high-dose MWCNT-A group developed lung tumors. Five rats in the low-dose MWCNT-B group developed lung tumors and seven rats in the high-dose MWCNT-B group developed lung tumors. In this study, crocidolite asbestos was also administered at a final dose of 1.0 mg per rat. Three rats in the crocidolite group developed lung tumors. These results indicate that thin tangled MWCNT fiber can induce lung cancer.

Saleh et al. (2022) administered double-walled CNTs (DWCNTs) to male F344/DuCrlCrlj rats by TIPS [65]. The rats were administered 15.625, 31.25, and 62.5 µg DWCNTs per rat every other day for 15 days. Eight administrations for total doses of 0.125 mg, 0.25 mg, and 0.5 mg per rat. The interim sacrifice was at 1 year, and the final sacrifice was at 2 years. At the interim sacrifice, DWCNT fibers were primarily found in granulation tissue, but in the 0.25 mg group five of seven rats examined had developed bronchiolo-alveolar hyperplasia and in the 0.5 mg group five of six rats examined had developed bronchiolo-alveolar hyperplasia (also see El-Gazzar et al. (2019) [66]). At the final sacrifice, 1 of 21 rats in the untreated group and 1 of 25 rats in the vehicle control group had developed lung tumors. In the 0.125 mg group 4 of 25 rats developed lung tumors; in the 0.25 mg group 4 of 26 rats developed lung tumors, and in the 0.5 mg group 7 of 24 rats developed lung tumors. Thus, in this study, the administration of DWCNT into the rat lung induced a significant increase in the development of lung tumors. In addition, one rat in the 0.25 mg group and one rat in the 0.5 mg group developed malignant pleural mesotheliomas. However, the mesotheliomas developed in the visceral pleura and there was no increase in HMGB1 levels in the pleural lavage fluid. Given that HMGB1 is released from mesothelial cells damaged by asbestos and that the released HMGB1 is involved in malignant transformation [67] (also see the Discussion in Saleh et al. [65]), it was concluded that the mesothelioma that developed in these rats was not relevant to human disease. In addition, one rat in the 0.125 mg group and one rat in the 0.5 mg group developed malignant pleural mesothelioma. However, male Fischer 344 rats develop spontaneous peritoneal mesotheliomas. Therefore, the single incidences of these lesions suggest that they were not treatment related. Overall, the studies by Saleh et al. (2020, 2022) [63,65] indicate that thin, tangled MWCNTs can induce tumors in the lungs of test animals, and in agreement with the Stanton and Pott hypotheses thin, tangled MWCNTs applied to the lung do not induce mesothelioma (Figure 2).

These studies indicate that MWCNT-7 induces malignant mesothelioma as well as lung cancer in rats. MWCNT-N, which is similar to MWCNT-7, also induced both lung cancer and malignant mesothelioma in rats. Notably, tangled agglomerates of MWCNTs were also carcinogenic in the rat lung. Importantly, in the workplace CNTs are usually found as agglomerates rather than single fibers [68]. Therefore, exposure of test animals to MWCNT agglomerates is relevant to human health.

**Table 5 nanomaterials-15-00168-t005:** Two-Year Carcinogenicity Studies.

Reference	Route of Administration	Animal	Administration/ Exposure Schedule	Terminal Sacrifice	CNT	Diameter Length	Dose per Animal	Lung Lesions
Sargent et al. (2014) [32]	Whole-body inhalation	B6C3F1 mice	Intraperitoneal Injection of methylcholanthrene (MCA) followed by inhalation exposure to 5 mg/m^3^ MWCNT-7 for 5 h/day for 15 days	17 months after exposure to MWCNT-7	MWCNT-7	Mitsui-7 MWNT-7 lot #061220-31	5 mg/m^3^	MWCNT-7 Promoted the development of MCA initiated lung cancer.
Kasai et al. (2016) [59]	Whole-body inhalation	F344/DuCrlCrlj rats	6 h/day 5 days per week 104 weeks	End of exposure at 104 weeks	MWCNT-7	92.9–98.2 ng 5.4–5.9 µm	Vehicle	Not carcinogenic
							0.02 mg/m^3^	Not carcinogenic
							0.2 mg/m^3^	Carcinogenic in the lungs of male rats No pleural lesions.
							2 mg/m^3^	Carcinogenic in the lungs of both male and female rats. Hyperplasia in the parietal pleura of both male and female rats.
Numano et al. (2019) [60]	Intratracheal Instillation	F344/DuCrlCr rats	125 µg once a week for 12 weeks (total dose of 1.5 mg/rat)	Lifetime study	MWCNT-7	Mitsui-7 MWNT-7	1.5 mg	1 rat died from a spontaneous pituitary tumor. The remaining 18 rats died from malignant pleural mesothelioma.
Hojo et al. (2022) [61]	Intratracheal Instillation	F344/DuCrlCrlj rats	1 administration of 0.125 or 0.5 mg/kg every 4 weeks. 26 administrations.	End of exposure at 2 years.	MWCNT-7	Mitsui-7, MWCNT-7 lot #060 125-01 k	Vehicle	1/30 Lung tumor 0/30 Malignant pleural mesothelioma
							3 mg/kg	3/29 Lung tumors 4/29 Malignant pleural mesothelioma
							12 mg/kg	11/28 Lung tumors 12/28 Malignant pleural mesothelioma
Suzui et al. (2016) [62]	Intratracheal Instillation	F344/Crj rats	1 administration of 125 µg/rat every other day. 8 administrations.	2 years	MWCNT-N		Vehicle	Not carcinogenic
						30–80 nm diameter Unfiltered: 4.2 ± 2.9 µm	1 mg/rat	4/12 Lung tumors, 3/12 malignant mesothelioma.
						30–80 nm diameter Flow through: 2.6 ± 2.9 µm	1 mg/rat	3/12 Lung tumors, 3/12 malignant mesothelioma.
						30–80 nm diameter Retained: Not determined	1 mg/rat	2/14 Lung tumors, 2/14 malignant mesothelioma.
Saleh et al. (2020) [63]	Intratracheal Instillation	F344/DuCrlCrlj rats	1 administration of 62.5 µg or 125 µg/rat per week. 8 administrations.	2 years			Vehicle	1/19 lung tumor No pleural lesions
					MWCNT-A	150 ± 43 nm 6.39 ± 3.07 µm	0.5 mg/rat	5/20 lung tumors No pleural lesions
							1 mg/rat	4/20 lung tumors No pleural lesions
					MWCNT-B	7.4 ± 2.7 nm 1.04 ± 0.71 µm	0.5 mg/rat	5/20 lung tumors No pleural lesions
							1 mg/rat	7/20 lung tumors No pleural lesions
Saleh et al. (2022) [65]	Intratracheal Instillation	F344/DuCrlCrlj rats	1 administration of 15.625 µg, 31.25 µg, or 62.5 µg/rat per week. 8 administrations.	2 years			Vehicle	1/21 Lung tumor No pleural lesions
					DWCNT	14.32 ± 10.04 nm Not determined	0.125 mg/rat	4/25 Lung tumors No pleural lesions
							0.25 mg/rat	4/26 Lung tumors No pleural lesions
							0.5 mg/rat	7/24 Lung tumors No pleural lesions
					MWCNT-7	76.49 ± 31.14 nm 8.79 ± 4.41 µm	0.5 mg/rat	3/9 Lung tumors (The other 16 rats died from malignant mesothelioma)

## 8. Conclusions

This review examines the studies that assess the toxicity and carcinogenicity of MWCNTs in the lungs of test animals and indicate that both thick, rigid MWCNTs and thin, flexible MWCNTs, including DWCNTs, induce inflammation and tumorigenesis in the lungs. Hojo et al. (2023) have determined that the lung carcinogenicity of MWCNT-7 is associated with persistent inflammation and ROS generation [69]. This agrees with the induction of inflammation in the lung by both thick rigid MWCNTs and thin flexible MWCNTs observed in short-term and medium-term studies, and the carcinogenicity of both thick rigid MWCNTs and thin flexible MWCNTs observed in 2-year studies. Another very important point is that MWCNTs are also reported to induce fibrosis in the lungs, and lung fibrosis is an important feature of pneumoconiosis, a serious disease that is known to be caused by inhalation of various dusts. Finally, as discussed by Heller et al. (2020) [70], the identification of CNTs as toxic substances does not indicate that these substances should be banned, but rather that measures be put into place so that these materials can be manufactured and used safely.

## Figures and Tables

**Figure 1 nanomaterials-15-00168-f001:**
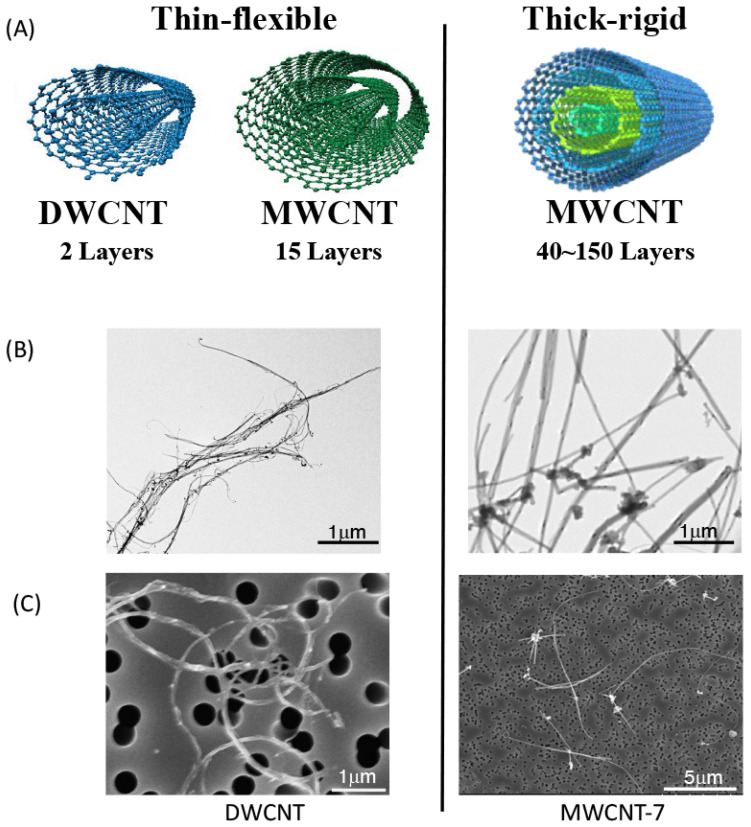
Structure of thin-flexible and thick-rigid MWCNTs. (**A**) Carbon nanotubes (CNTs) are cylindrical structures that consist of coaxially arranged graphene sheets. These sheets are generally referred to as “walls”. MWCNTs with fewer sheets are thin and flexible, and MWCNTs with more walls are thick and rigid. (**B**) TEM images of DWCNT and MWCNT-7 fibers. (**C**) SEM images of DWCNT and MWCNT-7 fibers.

**Figure 2 nanomaterials-15-00168-f002:**
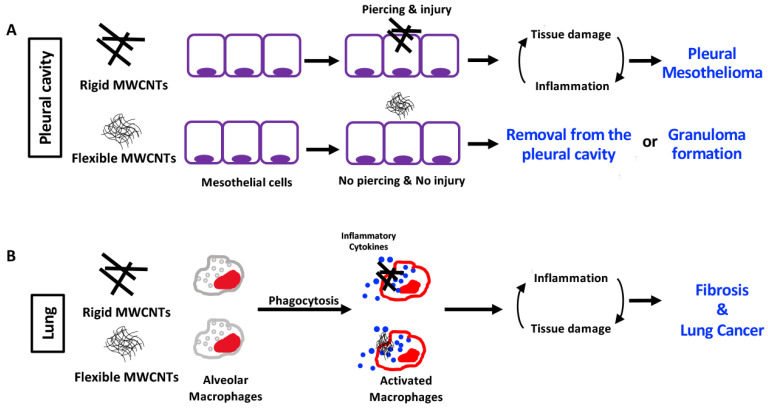
Carcinogenicity of thick, rigid and thin, tangled MWCNTs in the pleural cavity and lung. (**A**) In the pleural cavity, rigid MWCNTs pierce mesothelial cells, causing cell injury. The release of HMGB1 from injured cells promotes inflammation and tissue damage [26,67]. Persistent inflammation can lead to the development of mesothelioma. In contrast, thin- flexible MWCNTs do not pierce mesothelial cells, and thus there is no cell injury. Flexible fibers are removed from the pleural cavity or enclosed within granulomas. (**B**) In the lung, both thick, rigid and thin, tangled MWCNTs are engulfed by macrophages, leading to secretion of inflammatory cytokines and subsequent inflammation. Persistent inflammation is a key factor in the induction of fibrosis and tumorigenesis in the lung.

## Data Availability

No new data were created or analyzed in this study. Data sharing is not applicable to this article.

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
