# Peer review of "A Review of the Carcinogenic Potential of Thick Rigid and Thin Flexible Multi-Walled Carbon Nanotubes in the Lung"

_nanomaterials, 2025, doi:10.3390/nano15030168_

Round 1
Reviewer 1 Report
Comments and Suggestions for Authors
The review covers the recent advances on arcinogenic potential of thick rigid and thin flexible MWCNTs in the lung . The topic is of great interest to the researchers in the related fields. The manuscript can be published after some major issues being addressed.
1. Please add a typical morphology characterization diagram of carbon nanotubes .
2. Please list the animal models used in toxicology studies mentioned in the literature .
3. Please list the specific results of toxicological studies obtained from the literature and compare them, especially the conflicting results.
4. It is recommended to draw a diagram to illustrate the toxicological signaling pathways obtained from the literature .
5. More perspective content should be added to the summary.
Author Response
Please see the attached Reply to Reviewer 1.

Reviewer 2 Report
Comments and Suggestions for Authors
This article provides an overview of research on the carcinogenicity of multilayer carbon nanotubes (MWCNTs) in the context of the risk of lung diseases, including lung cancer. Despite the relevance of the topic related to the growing use of nanomaterials in industry and medicine, the article faces a number of methodological and structural problems affecting the perception of the results and their interpretation. This review examines the key shortcomings of the work, as well as offers recommendations for improving its quality and practical significance.
The manuscript should be reworked and submitted again.
1. The article is quite voluminous, but the structure of the review is not detailed enough: there are not enough sections and subsections that could facilitate the perception of the material and make the logic of presentation more convenient for the reader.
2. Some sections are overloaded with repetition of conclusions from various experiments, which makes it difficult to identify key aspects. For example, the repeated mention of research results on intraperitoneal injections creates a sense of duplication.
3. The author should have paid more attention to the actual applicability of the results to human health. For example, it is not entirely clear how the findings from animal experiments relate to potential risks for humans, given the differences in physiology.
4. Although the toxicological aspects of nanotubes are discussed, there is no detailed consideration of possible doses that may be critical for humans, especially with prolonged exposure.
5. The review presents an insufficient number of literature sources over the past 5 years (less than 20% of the total), which may indicate the use of outdated data and reduce the relevance of the information provided.
Recommendations for improvement:
1. Review the structure of the work, adding clear sections for introduction, methods, results, discussion, and conclusions.
2. Reduce the amount of repetition by combining duplicate sections and simplifying the presentation of information.
3. Include more context about the potential correlation between animal data and human diseases, based on epidemiological studies.
4. Add graphical elements (diagrams, tables) to visually represent the differences between the types of nanotubes and their effects.
5. Deepen risk analysis, taking into account real-world scenarios of human exposure at work or at home.
6. Complete the conclusions with more specific recommendations, for example, to propose strategies for minimizing risks.
Author Response
Please see the attached Reply to Reviewer 2.

Reviewer 3 Report
Comments and Suggestions for Authors
The manuscript entitled "A review of the carcinogenic potential of thick rigid and thin 1 flexible MWCNTs in the lung” provides a comprehensive overview of the carcinogenic potential of multi-walled carbon nanotubes (MWCNTs) in the lung, with a specific focus on the differences between thick rigid and thin flexible types. It is a very interesting topic in the field as biomaterial scientist rarely look at the long-term toxicity of CNTs, specifically carcinogenic properties. I would recommend the manuscript to be published after the minor issue below is properly solved:
1. It would be clearer if the author chose several key figures or tables as displayed items in this figure.
2. The conclusion of the carcinogenic trend of CNTs is clear given the comprehensive literature review that the authors collect. However, I am curious about detailed molecular mechanism of the carcinogenic and inflammation properties in lung. If the current research does not include the issue above, I suggest the authors include more perspective about it in the discussion.
Author Response
Please see the attached Reply to Reviewer 3.

Reviewer 4 Report
Comments and Suggestions for Authors
Review comments for “A review of the carcinogenic potential of thick rigid and thin flexible MWCNTs in the lung.”
Major comments:
1. Abstract and Introduction: The abstract effectively highlights the significance of reassessing the carcinogenicity of MWCNTs, but it would be beneficial to add some examples on the practical implications for workplace safety or public health.
2. Conclusions: The conclusions are comprehensive but the conclusion said, “measures be put into place so that these 571 materials can be manufactured and used safely.” Is not enough. It would be beneficial to add specific recommending next steps for regulatory guidelines or further research.
Minor comments:
1. Line 29: The sentence "Different types of CNTs vary in their diameter and length but in general they range from 10 to 200 nanometers in diameter, while their lengths generally range from a few micrometers tens of micrometers [1]" is slightly confusing. Consider revising to "Different types of CNTs vary in their diameter (10 to 200 nanometers) and length (a few to several tens of micrometers) [1]."
2. Grammatical Review: Sentences such as “Therefore, the carcinogenicity of thin MWCNTs is not associated with the carcinogenicity of individual thin MWCNTs, but rather to the carcinogenicity of MWCNT agglomerates” (Line 177-178) are repetitive and could be simplified for better readability. A thorough proofreading for redundancy and clarity is recommended.
Comments on the Quality of English Language1. Line 29: The sentence "Different types of CNTs vary in their diameter and length but in general they range from 10 to 200 nanometers in diameter, while their lengths generally range from a few micrometers tens of micrometers [1]" is slightly confusing. Consider revising to "Different types of CNTs vary in their diameter (10 to 200 nanometers) and length (a few to several tens of micrometers) [1]."1.
Grammatical Review: Sentences such as “Therefore, the carcinogenicity of thin MWCNTs is not associated with the carcinogenicity of individual thin MWCNTs, but rather to the carcinogenicity of MWCNT agglomerates” (Line 177-178) are repetitive and could be simplified for better readability. A thorough proofreading for redundancy and clarity is recommended.
Author Response
Please see the attached Reply to Reviewer 4.

Round 2
Reviewer 1 Report
Comments and Suggestions for Authors
accept
Reviewer 2 Report
Comments and Suggestions for Authors
The manuscript may be accepted in its current form.